# Outcomes of Device Closure of Atrial Septal Defects

**DOI:** 10.3390/children7090111

**Published:** 2020-08-25

**Authors:** P. Syamasundar Rao

**Affiliations:** University of Texas at Houston McGovern Medical School, Children’s Memorial Hermann Hospital, Houston, TX 77030, USA; P.Syamasundar.Rao@uth.tmc.edu

**Keywords:** atrial septal defect, percutaneous closure, outcome, transcatheter occlusion, King and Mill’s Device, Rashkind’s devices, clamshell occluder, buttoned device, Atrial Septal Defect Occluding System, Das Angel Wing Device, Amplatzer Septal Occluder, Amplatzer Cribriform Device, Gore HELEX^®^ Device, GORE^®^ CARDIOFORM ASD Occluder

## Abstract

Several devices have been designed and tried over the years to percutaneously close atrial septal defects (ASDs). Most of the devices were first experimented in animal models with subsequent clinical testing in human subjects. Some devices were discontinued or withdrawn from further clinical use for varied reasons and other devices received Food and Drug Administration (FDA) approval with consequent continued usage. The outcomes of both discontinued and currently used devices was presented in some detail. The results of device implantation are generally good when appropriate care and precautions are undertaken. At this time, Amplatzer Septal Occluder is most frequently utilized device for occlusion of secundum ASD around the world.

## 1. Introduction

Following the first report of closure of atrial septal defects (ASD) surgically by Kirklin and his associates in the mid-1950s [1], it quickly became a regular treatment for ASDs [1,2,3]. The usual management of option of moderate and large ASDs until recent times is correction by surgery. While closure by surgery of these defects is safe and successful with negligible mortality [4,5], the morbidity related to sternotomy/thoracotomy cannot be circumvented. As a result, extensive attempts were undertaken by cardiologists around the world to develop a percutaneous method without surgery for ASD closure. Since the initial reports in the mid-1970s by King [6,7,8], Rashkind [9,10,11] and their associates of ASD occluding devices, several types of devices, as appraised in the author’s prior publications [12,13,14], have emerged. Accordingly, two methods, namely, surgical and transcatheter closure, are now available for ASD occlusion.

While there are a few studies comparing surgical with device closure, the existing studies [15,16] indicate similar effectiveness. Yet, the percutaneous occlusion is less invasive and needs no cardio-pulmonary bypass. In addition, the percutaneous occlusion seems to require shorter stay in the hospital (1 day vs. 4.3 days), has less prevalence of complications (10% vs. 31%) and is with reduced expense (US $11,000 vs. $21,000) [17]. Other drawbacks of surgical intervention are the residual scar and emotional trauma to the children and/or their parents. The overall assessment is that the device closure methods were demonstrated to be safe, cost-effective and satisfactorily measure up to closure by surgery [15,16,17,18]. Percutaneous device occlusion of ASDs is now a recognized treatment in the majority of heart centers offering state of the art care to subjects with congenital heart defects (CHDs) [18,19,20,21,22]. At this time, ASD closure by surgery is mostly offered for subjects whose ASDs have deficient rims of the atrial septum in whom the defect is hard to occlude with percutaneous methods or has had attempts to transcatheter occlude the defect without success. Furthermore, in case surgical repair of other CHDs is planned, closure of ASD by surgery can be undertaken at that time.

Despite the description of many different devices [12,13,14], only four devices, namely, the Amplatzer Septal Occluder (ASO) (St. Jude Medical, Inc., St. Paul, MN, USA—Abbott), Amplatzer Cribriform Device (St. Jude Medical, Inc.), Gore HELEX^®^ Device (W.L. Gore, Flagstaff, AZ, USA) and GORE^®^ CARDIOFORM ASD Occluder (W.L. Gore), are currently approved for occlusion of ostium secundum ASDs by the US Food and Drug Administration (FDA) in the USA. Occlutech Septal Occluder which is akin to that of ASO is utilized outside the USA.

The purpose of this review is to present outcomes of device occlusion of ASDs. The outcome of the some of the devices which were either discontinued or withdrawn from further clinical use for varied reasons will be reviewed first. The devices not yet approved by the FDA will be mentioned and then, the devices currently in use and approved by FDA will be reviewed. Most of the devices were first tried in experimental animals with subsequent studies in children and adults. Observations of experimental animal models, when present will be reviewed first followed by results of human clinical trials.

## 2. Discontinued Devices

### 2.1. King and Mill’s Device

King and his colleagues were able to close ASDs via percutaneously implanted devices (Figure 1); they first reported their results in the mid-1970s [6,7,8]. King & Mills [6] initially implanted the umbrella devices across ASDs produced surgically in adult canines. The device implantation was attempted in nine dogs and the procedure was successful in five dogs. Full occlusion of the ASD and endothelialization of the devices placed across the ASD was demonstrated at the follow-up examination [6]. Following testing in animal models [6], human trials began [7,8]. Evaluation in the catheterization laboratory was undertaken in 18 patients and ten (56%) of these patients were assessed to be appropriate patients for transcatheter occlusion. Device placement was a success in five (50%) patients. The patient’s ages varied between 17 and 75 years; the median age was 24 years. The balloon stretched diameter of the ASD varied between 18 and 26 mm. Four patients had ostium secundum ASDs with left-to-right shunt while the fifth patient had had a stroke secondary to a presumed paradoxical embolism. In patients with ostium secundum ASD, there was improvement of symptoms along with decreased heart size. Follow-up cardiac catheterization data did not demonstrate shunts by oximetry but minor shunts were seen by hydrogen curves, a highly sensitive method to detect small shunts. A follow up study 27 years later [23] revealed that 4 patients were well with evidence of occluded ASDs and without any device-related adverse events. However, one patient expired secondary to Hodgkin’s disease and a stroke nine years following device occlusion [23]. While these results were assessed to be satisfactory, neither King and his associates nor other investigators have utilized this procedure subsequently. While the reason for this is not clear, it likely to be due to large size (23F) of the sheath required to implant the device and complex maneuvering necessary for delivering the prosthesis across the ASD.

### 2.2. Rashkind’s Devices

Rashkind and colleagues designed a somewhat different type of ASD occluder (Figure 2) [25]. Studies in occluding ASDs produced surgically in canine and calf models demonstrated the feasibility of the technique. During follow-up, endothelialization of the umbrella components of the device were seen [9]. After successful results in experimental models, the investigation of ASD occlusion was extended to human subjects [10,11]. Rashkind selected 33 patients for the human investigation. In ten patients no attempts were made to insert the umbrella devices because the ASDs were either large (in six patients) to safely implant the device or the ASDs were very small (in four patients) to warrant occlusion. Of the rest of the 23 patients, 14 (61%) underwent satisfactory ASD occlusion and in the remaining 9 (39%), the outcome was not satisfactory. The first six device deployments were with three-rib prostheses and the rest were six-ribbed umbrellas. The size of the sheath required for implantation of these devices was 16F; an improvement from 23F sheath requirement for the King’s device. While satisfactory outcomes were accomplished in half (50%) of the patients, multiple issues were found; these include, the need for a large delivery sheath (16 F) for device placement, doubts about complete endothelialization of the prosthesis and problems associated with freeing up the device should the hooks of the prosthesis get entangled onto the mitral apparatus or wall of the left atrium. Consequently, the device was redesigned by Rashkind as a double disc umbrella in a manner similar to a patent ductus arteriosus closure device that was being developed simultaneously [11,26]. The double disc device was deployed satisfactorily across ASDs (Figure 3) in experimental animals [11].

### 2.3. Clamshell Occluder

When Rashkind’s double disc prosthesis was utilized by other investigators, they found that the umbrellas do not re-position against the atrial septum [27] and therefore, a modification of the device was undertaken by inserting a second spring in the middle of each of the arms of the device; this modified device was named Clamshell Occluder (Figure 4) [27]. Initially, the clamshell occluder was utilized to close surgically produced ASDs in lambs. The device was successfully implanted in six of the eight lambs. In the remaining two lambs the device embolized. Full closure of the atrial defects was observed in four lambs. At follow-up at one and two months after device placement complete endothelialization of the components of the device was shown [27]. In the clinical trial that followed, cardiac catheterization was performed in 40 patients with intent to occlude their ASDs. The device was successfully implanted in 34 (85%) patients [28]. They used 11F sheaths to implant the devices. They encountered one major complication, namely death because of cerebral embolus, thought to be due to iliac vein thrombus dislodgment while the delivery sheath is positioned. Otherwise, the procedures were uncomplicated. However, two (6%) devices embolized into the descending aorta at iliac bifurcation. Transcatheter retrieval of the embolized devices was undertaken followed by surgical correction of the ASD electively. Twelve (63%) of 19 had no residual shunts on echocardiographic studies during follow-up [28]. Studies by these and other cardiologists, as reviewed elsewhere [14] were conducted. But, several investigators [29,30,31] detected fractures in the arms of the clamshell occluder in 40% to 84% of the patients with infrequent embolization [29,30,31]. Presumably because of these complications additional human trials with the clamshell occluder were stopped by the FDA and the investigators.

### 2.4. Buttoned Device

Sideris and his associates [32,33] developed a “buttoned device” in 1990. The buttoned device was first tried in experimentally produced ASDs in piglets. Successful device implantation was accomplished in 17 (85%) of 20 attempts [32]. Complete closure of the atrial defects along with endothelial covering of the device was shown in all the 17 button device placements [32]. Following the demonstration of feasibility, safety and effectiveness in experimental animals [32], studies in both children and adults began; single institutional [33,34,35,36,37,38,39,40], US FDA-approved [41,42] and international [43,44,45] clinical trials were conducted. The device requires 8F or 9F sheaths for device implantation. The device has also undergone a number of modifications to improve its performance [35,43,44,45,46]. 

A total of 768 patients had buttoned device implantation to occlude ostium secundum ASDs from 1989 to 2002 at 40 institutions around the world [47]. For this review, the patients were divided into three cohorts on the basis of the type of buttoned device implanted. Cohort I consisted of 180 patients [43] in whom single-button (first-, second- & third-generation) (Figure 5 and Figure 6) buttoned device implantations were performed between 1990 and 1993. Cohort II is made up of 423 patients [45] in whom double-button (fourth-generation) buttoned device (Figure 6 and Figure 7) placements were performed between 1993 and 1997. Cohort III is composed of 175 patients [24,46] in whom centering-on-demand (fourth-generation with centering mechanism—Figure 8) buttoned device implantations were performed from 1999 to 2002. The immediate results showed that the device implantation feasibility was similar (90% vs. 89%; *p* > 0.1) for Cohorts I and II but was higher (97%; *p* < 0.05 to <0.01) for Cohort III. The number of successful device implants relative to the number implanted was better (*p* < 0.001) for Cohorts II (99%) and III (100%) than for Cohort I (92%). The total major complication rate for Cohort I (7.8%) was higher (*p* < 0.01) than for Cohort II (1.4%) and Cohort III (0.6%). Similarly, the unbuttoning rate for Cohort I (7.2%) was higher (*p* < 0.01) than for Cohort II (0.4%) and Cohort III (0.6%). However, the rates of effective occlusion (trivial or no residual shunt) for Cohorts I (92%), II (90%) and III (95%) were similar (>0.05 to >0.1). The duration of follow-up for Cohort I (46 ± 20 months) was longer than that for Cohort II (23 ± 15 months; *p* < 0.002) and Cohort III (12 ± 9 months; *p* < 0.01). The re-intervention rate for Cohort I (8%) was higher than that for Cohort II (5%; *p* < 0.02) and Cohort III (0.6%; *p* < 0.01). The actuarial re-intervention-free rates for Cohort I (93.5%, 92.1% & 89.9% at 1, 2 and 5 years, respectively) were similar (*p* > 0. 1) to those for Cohort II (95.3%, 92.6% & 90.6% at 1, 2 and 5 years, respectively) but were lower (*p* < 0.05) than for Cohort III (99% and 99% at 1 and 2 years, respectively).

In summary, while the patients in Cohort III, who had the centering-on-demand (COD) buttoned device closure, were similar to the other two cohorts in terms of the size of the ASD, the implantation feasibility improved (*p* < 0.01), the effective occlusion rates remained stable (*p* > 0.05) and unbuttoning was abolished. While the re-intervention-free rates were higher in Cohort III, the follow-up duration was shorter than that of the other two cohorts. These data also confirm our hypothesis that the modifications of the device improve the device’s performance [24,47].

Following the completion of FDA-sponsored and international clinical trials, Dr. Sideris, who invented the buttoned device decided not to seek pre-market approval (PMA) from the FDA and, therefore the buttoned device is no longer available for clinical use [24].

### 2.5. Atrial Septal Defect Occluding System (ASDOS)

Babic and colleagues [48,49] designed ASDOS (atrial septal defect occluding system) (Figure 9) in 1991. The ASDOS device was used to occlude inter-atrial defects enlarged by balloon dilatation in 20 pigs [48,49,50,51]. Complete coverage of the device components with smooth, scar-like tissue was found by three months after the procedure. 11-F long sheath is required for implantation of the ASDOS. 

Clinical trials in children [52], in adults [48,53] and in 20 European hospitals [54] were undertaken. Of the 800 patients screened, 350 (44%) patients were selected for device implantation; of these, 318 (91%) had successful implantation. ASDOS devices were implanted via 11F sheaths. The failure rate was 9% (N = 32). Transcatheter retrieval was accomplished in 26 (7%) and surgical retrieval was elected in six (2%). Other complication observed were: device embolization in 3 (0.9%) patients, thromboemboli in 3 (0.9%), cardiac perforations in 6 (1.6%) and suspected infections in 2 (0.6%). Surgical removal of the device was necessary in 3% (N = 11), frame fractures were seen in 20% of patients and formation of thrombus occurred in 25% during follow-up. As a result of the high complication rate, the inventor of ASDOS abandoned the device [48] and the device is not in use at present. The device was modified by placing a stent between the umbrellas to afford good centering of the device; however, this modified device is not available for human use [48].

### 2.6. Das Angel Wing Device

Das and associates [55] designed a self-centering device in 1993; the device was named Das Angel Wing Device (Figure 10) and can be delivered transvenously via an 11F sheath. Percutaneous closure with this device was attempted in surgically produced ASDs in 20 adult canines. The device was successfully implanted in 19 (95%) of 20 canines. Angiograms subsequent to the device occlusion demonstrated trivial shunts in 2 dogs and no residual shunts in 17 canines. Follow-up (2 to 8 months) studies in six canines demonstrated that the trivial shunt that was present in one dog immediately following closure had disappeared on repeat study. No embolization of the devices was observed either at the time of device implantation or during follow-up. Microscopic examination revealed covering of the prosthesis with smooth endocardium, enmeshed in mature collagen tissue in 3 dogs at 8 weeks following device placement. Das and colleagues concluded that the results in the dog experiments support the device use in human subjects because of the self-centering feature of the device and safe and effective ASD occlusion rates in the canine experiments [55].

Studies in human subjects were conducted within [56,57,58] and outside [59] the US. Fifty patients with ostium secundum ASD were enrolled in the Phase I trial; device implantation was successful in 46 (92%) patients [56,58]. Device mal-position occurred in four (8%) patients which required removal of the device at surgery; the ASD was closed at the time of surgery. There were also other complications related to the procedure in three (6%) patients. Results of follow-up transesophageal echocardiography (TEE) examination in 34 patients were reviewed; no residual shunt was seen in 31 (91%) patients. Another fifty patients with secundum ASD were enrolled in a multicenter, prospective, non-randomized study [59]; the procedure was successful in 47 (94%) patients; the remaining three (6%) patients required surgery. Multiple other complications (transient complete heart block in three patients, lack of non-deployment of right atrial disc in three, blood clots in 2, hemopericardium with tamponade in one and large residual shunt in one) occurred. Residual shunts were seen in 27% patients immediately after the procedure while at 1 to 17 month follow-up residual shunts were seen in only three patients. The authors concluded that percutaneous occlusion of ASD with Angel-Wing’s device is feasible and effective in both children and adults and that design modifications were required to avoid serious complications associated with device implantation [59]. Forty-seven patients were enrolled in the Phase II clinical trial; the results in this trial were essentially similar that seen in the Phase I trial [58].

Before the conclusion of the Phase II trial, the study was discontinued in order to reconfigure the device [58]. The modified device, named Guardian Angel Wing (Angel Wing II) was composed of rounded (instead of square shape) right and left atrial discs. It was redesigned to be easily retrievable and repositionable yet maintain the self-centering characteristics. While it was mentioned that the redesigned device will go into clinical trials shortly thereafter [58], no further reports of either the Angel Wing or Guardian Angel devices appeared in the literature and it appears that the device was shelved.

## 3. Devices Not Yet Approved by FDA

There are a number of devices which have not received approval by the FDA for unrestricted clinical application namely, modified Rashkind PDA umbrella device, CardioSEAL device, STARFlex device, Sideris’ wireless devices (including trans-catheter patch), hybrid buttoned device, BioSTAR, BioTREK, Occlutech, Cardia devices (ATRIASEPT I-ASD device, ATRIASEPT II-ASD and ULTRASEPT), Solysafe Septal Occluder device and PFM ASD-R device. Discussion of these devices is beyond the scope of this presentation but the information pertaining to these devices is briefly reviewed elsewhere [14] for the interested reader.

## 4. Devices Currently in Use

### 4.1. Amplatzer Septal Occluder

Amplatz and his associates [60] designed Amplatzer Septal Occluder (ASO) (Figure 11A), which was utilized to transcatheter occlude ASDs created surgically (10 to 16 mm in diameter) in 15 mini-pigs. ASO was successfully deployed in 12 (80%) of these animal models. Angiography demonstrated full occlusion of the ASD in 7 out of 12 subjects shortly following implantation of the device and in 11 of 12 pigs seven days later. Neo-endothelialization with incorporation of the device into the fibrous tissue was observed by three months. It was concluded that small introducing system to implant the device and favorable experimental results indicate potential for clinical trials in human subjects [60].

The first clinical study with the ASO was reported by Masura [61]. The study subjects included thirty patients, aged 2.9 to 62.4 years (median 6.1 years) with weights of 13 to 69 kg (median 22 kg). Transcatheter occlusion via 7F sheaths with the ASO was performed in ASDs measuring 5 to 21 mm (median 12.5 mm) by TEE and 7 to 19 mm (median—14 mm) by balloon-stretch. The Qp:Qs prior to occlusion was 2.3 ± 0.6. After device occlusion with ASO, complete closure in 17 (57%), trivial shunts in 10 (33%) and moderate shunts in 3 (10%) out of 30 patients was observed. On the day after the procedure complete closure was seen in 24 (80%) of 30 patients. Follow-up at a median of six months continues to show good results. Masura and associates concluded that ASO occlusion of ASDs is associated with safety and effectiveness in producing full occlusion of secundum ASDs up to a diameter of 21 mm [61]. Subsequently other investigations in children [62,63,64,65,66,67,68,69,70] and adults [61,64,71,72,73,74,75,76,77,78,79] including FDA-approved US trial [80] and multi-institutional European studies [70,81], as reviewed and tabulated elsewhere [20,21,24,82], demonstrated excellent results. These studies confirmed encouraging results with full occlusion rates of 62 to 96% at device implantation, which increased further at 6- to 12-month follow-up evaluation to 83 to 99% [59,63,64,65,66,67,68,69,70,71,72,73,74,75,76,77,78,79,80,81,83].

At our institution, we performed the occlusion of more than 300 ostium secundum ASDs with the ASO. The results of the first 65 patients were reported in a poster format [84]. Cardiac catheterization was performed in 65 patients with the objective to occlude their ASDs with the ASO during years 2003 to 2011. Devices could not be implanted in 4 (6%) patients secondary to flail or inadequate septal rims. Transcatheter occlusions were successfully performed with the ASO in the remaining 61 (94%) patients.

The patients’ ages varied between 3.1 and 17 years (mean—7.9 years). The weights varied from 11.8 and 69 kg (mean—29.6 kg). Twenty-four were boys and 42 were girls. The mean Qp:Qs prior to occlusion was 1.8:1.0 with a range of 0.9 to 4.5. The diameters of the atrial defects by TEE ranged from 5.6 to 29.5 mm (mean—14.0 mm) and by left atrial angiography they measured between 5.8 and 25.9 mm (mean—14.5 mm). The balloon-stretched ASD diameter was between 10.2 and 25.8 mm (mean of 17.9 mm). The ASDs were classified as complex in 11 (17%) patients (defect diameter >26 mm in five patients, 2 or more ASDs were present in five patients and the atrial septum was fenestrated in one patient). The PA systolic pressures ranged between 11 and 34 mmHg with a mean of 20 mmHg. The implanted device sizes varied between 8 and 34 mm. The most commonly utilized device size was the 20 mm ASO. The device was successfully deployed in 61 (100%) of 61 patients in whom the ASO was implanted or in 61 (93.8%) of 65 patients who had catheterization with the objective of ASD closure. No device dislodgements occurred. An example of device occlusion of ASD with ASO is shown in Figure 12. Echo-Doppler studies performed on the day after ASD closure revealed small shunts in six patients and trivial shunts in four patients, as defined in our prior study [42]. Effective occlusion, classified as an absent (*n* = 51) or trivial (*n* = 4) residual shunt [42] on echocardiographic studies carried out on the day after ASD closure, was demonstrated in 55 patients (90%). Mild mitral insufficiency was seen by color flow imaging in two patients, which was transient and resolved spontaneously 15 months later. The PR interval on the electrocardiogram (ECG) performed on the day after ASD occlusion varied between 106 to 210 msec with a mean of 146 msec. There was evidence for first-degree heart block in seven patients and complete heart block in one. No other complications were seen.

Follow-up data was available in 100% patients ranging from one month to 78 months with a mean of 26 months. During this period, one (1.6%) patient with complete heart block required pacemaker implant. No other interventions were required. The PR interval on the ECG performed six months after ASD occlusion varied between 100 and 204 msec with a mean of 137 msec. There was evidence for prolonged PR interval in three patients, second-degree heart block in one patient and complete heart block in one patient (the same patient described in the immediate results section above). Echocardiographic studies demonstrated good position of the ASO. There was continuing decrease and resolution of the remaining shunts. At six-month follow-up the residual shunts were trivial in two children and small in five. One year following device implantation, a trivial residual shunt was seen in one child. No residual shunts were seen at and beyond 15 months after ASD closure. Paradoxical embolism did not recur in the two patients who had their ASD/PFO occluded to prevent additional incidents of cerebrovascular accidents (CVAs)/transient ischemic attacks (TIAs). Deaths have not occurred during the whole study period [47,84].

The majority of the studies, including our own detected minimal complications. Nonetheless, migration of the device with perforation of the aortic wall by the ASO, with some patients developing fistulous connection between the aorta and the right or left atrium [85,86,87] in 1 in 1000 ASO implantations was observed. Detailed analysis of this issue [87,88] concluded that this complication is probably due to over-sizing of the ASO (Figure 13). Accordingly, it was suggested that the size of the device should not exceed the ASD diameter by 1.5 times [87,88]. In over 300 ASO implantation at the author’s institution over the last 18 years, we have not seen this complication despite diligent and methodical follow-up echocardiographic studies; this is presumed to be related avoiding over-sizing at our institution.

The ASO has become the device of preference in the USA on the basis of the simplicity of device implantation, retrieval and repositionability (when needed) and that it received FDA approval.

#### Advantages and Limitations

The advantages of ASO are delivery via small sheaths, user-friendly characteristics, ability to retrieve and re-position the device with ease and that it can be used to occlude small, medium and large ASDs. The disadvantages are potential for producing fistulous connection between the aorta and the atria [85,86,87], although this is largely related to use of over-sized devices [87,88].

### 4.2. Amplatzer Cribriform Device

The Amplatzer Cribriform Device (Figure 11D) was designed to occlude fenestrated atrial defects [89]. Hijazi and Cao [89] when they described the device showed the usefulness of Amplatzer Cribriform Device in occluding fenestrated ASDs. Zanchetta and colleagues [90] used Amplatzer cribriform occluders in 13 patients and showed that complete closure occurred in 67% patients in 24 h, 79% in one month and 96% in 2 years after the cribriform device occlusion. In a subsequent study by Numan et al. [91], thirteen of sixteen patients had successful occlusion. Full occlusion was shown in 77% of patients on the day following the procedure. 92% of patients had a complete occlusion of the defects at six and 12 months after the procedure [91]. Other investigators documented similar encouraging results [92,93]. The author’s personal experience with this device was limited but successful (Figure 14 and Figure 15). The Amplatzer Cribriform Device has also been employed, mostly on off-label basis to successfully occlude patent foramen ovale (PFO) [94,95,96,97,98,99] before the approval of Amplatzer PFO Occluder (Figure 11B) by the FDA for closure of PFOs.

#### Advantages and Limitations

The Amplatzer Cribriform Device is easy to deploy and re-position as needed, similar to ASO and is useful in occluding fenestrated ASDs. However, all the defects must be covered by the discs of the device to avoid residual shunts.

### 4.3. Gore HELEX^®^ Device

Gore HELEX^®^ Device (Figure 16) occlusions were undertaken in 24 dogs who had ASDs created surgically. The device deployment rate was 100%. The rate of complete occlusion at device implantation was 88% by TEE which improved to l00% two weeks later [100,101]. These investigations also showed that neo-endothelialization and covering of the device with fibrous connective tissue usually occurs by three months. Following successful implantation feasibility in experimental animals [100,101], testing in children and adults commenced [102,103,104,105,106].

The first clinical implant was performed in 1999 [102,103] and the FDA Phase I feasibility study began in 2001. In this FDA-sponsored study [104], the device was successfully implanted across the ASD in 119 (88.1%) of the 135 patients with the intent to occlude. Successful defect closure with full occlusion or trivial shunt was seen by echocardiography at one year follow-up in 98% patients while the remaining 2% had significant residual shunts. An example of device occlusion of ASD with Gore HELEX^®^ device is shown in Figure 17.

Wire frame fractures of the device were noted in 5% patients. Adverse events such as device embolization/malfunction needing device removal in 5 and retroperitoneal hemorrhage in 1 and acute, confusional migraine in 1 were observed in 5.9% (7 of 119) of device implantations. The results of 119 Gore HELEX device ASD occlusions were compared with those of 128 surgical ASD closures [104]. The groups were thought to be statistically comparable. Successful closure rates, major and minor adverse events rates at the time of the procedure and adverse event rates at 12 month follow-up were similar for both groups. The authors’ conclusions were that the HELEX septal device occlusion of ASD is associated with safety and effectiveness [104]. However, the device closure cohort had shorter anesthesia time and lesser duration of hospitalization than for the surgical group.

Subsequent review of 435 patients who had Gore HELEX^®^ device occlusion in the USA (which included feasibility, pivotal, continued access and post-approval cases) was undertaken [105]. The results demonstrated that 93% of patients had successful clinical result at follow-up in 12-months. Freedom from major adverse events was 95% at one year after device occlusion. The authors stated that this is the largest cohort of Gore HELEX^®^ Device occlusions and the results reiterate that the Gore HELEX^®^ Device occlusion of ostium secundum ASDs is safe and effective. The complexity of device deployment and non-conformity of the device across the ASD were observed in some of the studies. In addition, the Gore HELEX^®^ Device is commonly viewed as an appropriate device for the occlusion of small to medium-sized ASDs. 

#### Advantages and Limitations

The Gore HELEX^®^ Device is found to be useful in occluding small ASDs. The complexity of device implantation, inability to conform to the atrial septum in some patients and not being useful for occluding all sizes of ASDs are its limitations.

### 4.4. GORE^®^ CARDIOFORM ASD Occluder

Among the initial reports of the clinical use of GORE^®^ CARDIOFORM ASD Occluder (GCO) (Figure 18) was a study encompassing 22 patients with ASD, aged 25.8 ± 4.6 years [106]. The size of the ASDs was 11.4 ± 0.5 mm by TEE or intracardiac echocardiography (ICE). GCO implantation was successful in all 22 patients. No residual shunts were seen both immediately following device occlusion and at follow-up evaluation [106]. Symptomatic improvement and normalization of right ventricular dimensions were seen. The authors concluded that GCO is an efficient device for closing ASDs up to 15 mm [106]. An example of device occlusion of ASD with GCO is shown in Figure 19.

Other single institutional [107,108] and Canadian [109], European [110,111] and US [112] multi-institutional studies indicated feasibility, safety and efficacy of GCO in occluding secundum ASDs of small to moderate size [107,108,109,110,111,112].

Fifty pivotal and 350 continued access patients with ASD have undergone GCO occlusion between 2012 and 2015 at 21 institutions in the US [112]. The median age at the time of device occlusion was 6.9 years and the median weight was 25 kg. The ASD size by TEE or ICE was 9.7 ± 3.1 mm and 12.0 ± 3.1 mm by balloon stop flow. GCO was successfully implanted in 374 (93.5%) of the 400 patients with intent to occlude. The most frequent reasons for non-implantation of GCO were inability to seat the device across the defect, prolapse of the device or significant residual shunt. Most commonly implanted device sizes were 25mm and 30mm GCOs. After device deployment full occlusion was demonstrated in 77% patients which improved to 86% at six months and 93% at 36 months. Residual shunts decreased with time so that only 0.4% patients had significant residual shunt 36 months after device implantation. Successful clinical results were observed in 90.2% patients at the time of device implantation and in 98.8% patients 6 months later. No serious adverse events occurred in 98.3% patients for 30 days following the device closure. No embolization of the devices were seen nor a need for re-intervention arose during six months after device deployment. The study authors concluded that GCO has excellent clinical results and safety profile and that it is useful in occluding defects with balloon-stretched diameter ≤17 mm [112]. 

#### Advantages and Limitations

The GCO is the latest of the FDA-approved ASD-occluding devices; there was a remarkable improvement in complexity of device deployment compared to the Gore HELEX^®^ Device. But non-conformability to the atrial septum in some patients and non-applicability to large ASDs are its drawbacks.

## 5. Discussion

Ostium secundum ASD is a common CHD both in children and adults. ASDs result in left to right shunt which causes dilatation of the right heart (right atrium, RV and main and branch PAs). Pulmonary vascular obstructive disease is not usually seen until late adulthood. The majority of patients are asymptomatic and are detected because of asymptomatic murmurs or findings on echocardiograms performed for other reasons; rarely some patients may present because of easy fatigability, arrhythmia and even more rarely with signs of heart failure. The ASDs can easily be diagnosed by the clinical features, chest X-ray and ECG and quantified by echo-Doppler studies. Surgical and more recently, transcatheter closure is generally recommended, usually performed around ages 3 to 5 years. Patients presenting later are addressed as and when they are identified. The indications for ASD closure are right heart volume overload manifested by dilatation of right atrium and RV with flat or paradoxical inter-ventricular septal motion demonstrated during echocardiographic studies and/or Qp:Qs > 1.5:1.0 during a invasive hemodynamic study. While the majority of the reports which compared surgical with device occlusion indicate that both are equally effective, device occlusion is non-invasive, has fewer complications, requires shorter hospital stay and is less expensive [4,5,15,16,17,18]. The device closure is now a well-known practice in the majority of institutions around the world that provide the state of the art cardiac care [18,19,20,21,22]. Surgery is used for patients with deficient septal rims and in patients in whom the device closure was unsuccessful. Several devices have been described and experimented over the years and currently, only four devices have received the FDA approval for clinical use. The outcomes of devices both in the animal experimental models and in human clinical studies were reviewed. The results are generally good when appropriate care and precautions are taken. A comparison of the FDA-approved devices for ostium secundum ASDs is presented in Table 1 and Table 2.

Currently, Amplatzer Septal Occluder is the most frequently utilized device for occlusion of secundum ASD around the world.

## Figures and Tables

**Figure 1 children-07-00111-f001:**
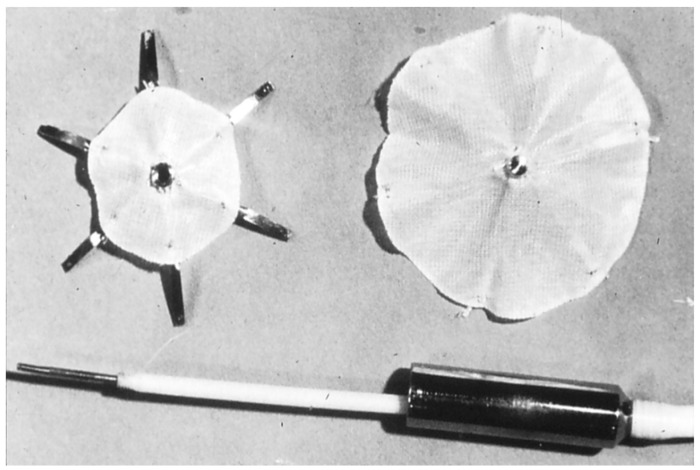
Photograph of King and Mill’s device. Reproduced from reference [24].

**Figure 2 children-07-00111-f002:**
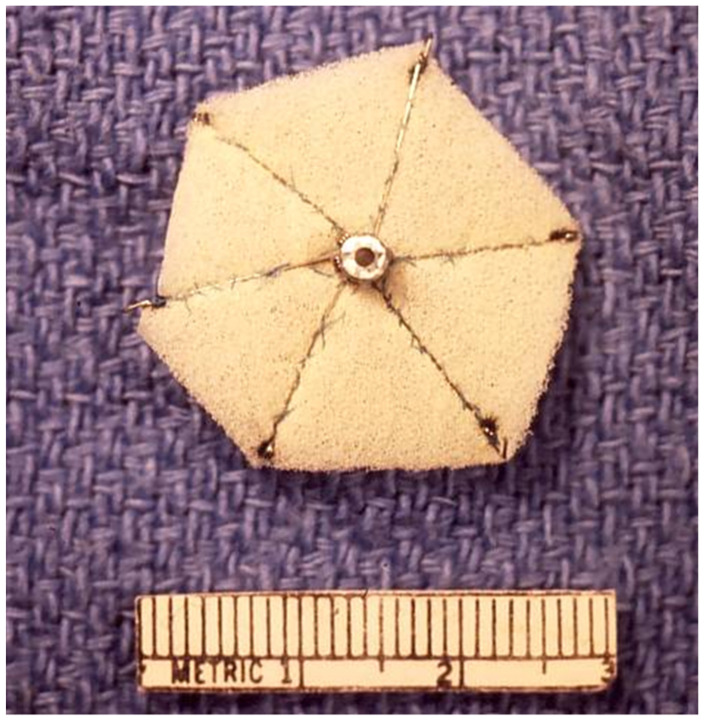
Photograph of Rashkind’s hooked device. Reproduced from reference [24].

**Figure 3 children-07-00111-f003:**
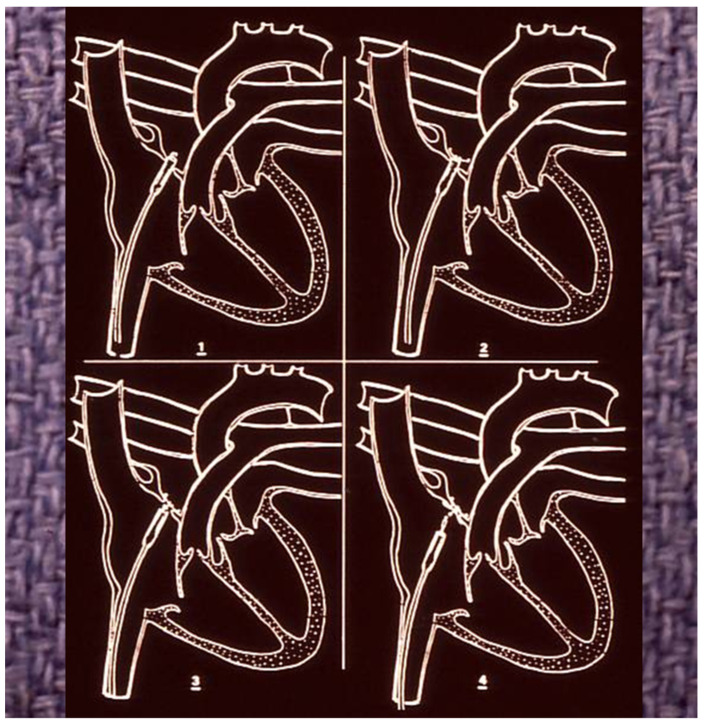
Method of implantation of Rashkind’s double disc device. Reproduced from reference [24].

**Figure 4 children-07-00111-f004:**
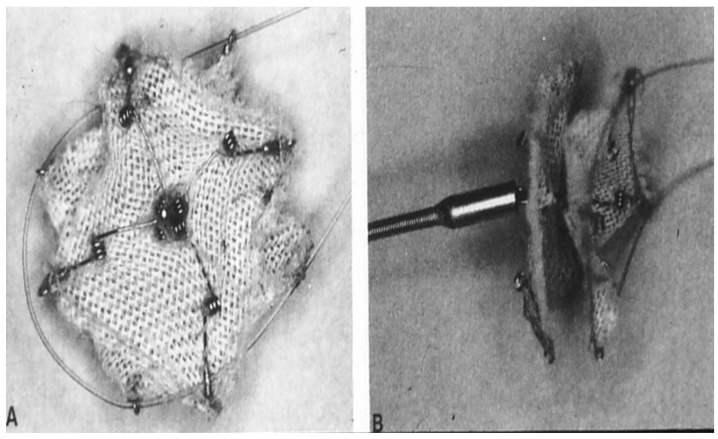
Photograph of a Clamshell device (**A**,**B**). Reproduced from reference [24].

**Figure 5 children-07-00111-f005:**
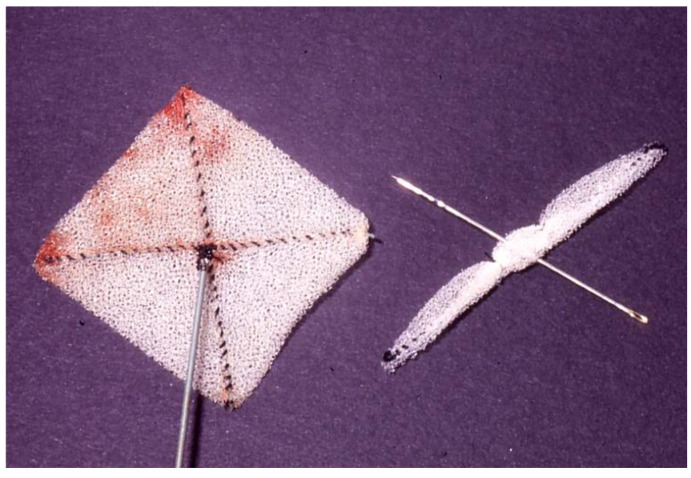
Photograph of the original buttoned device. Reproduced from reference [24].

**Figure 6 children-07-00111-f006:**
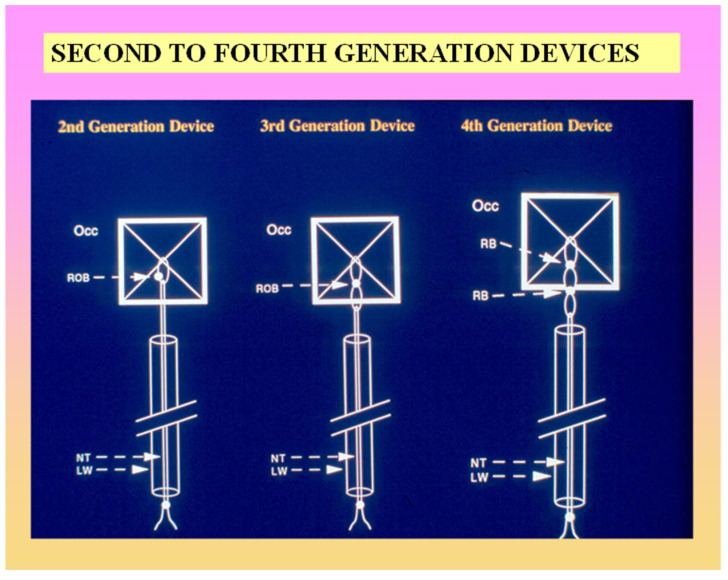
Cartoon depicting various components of several generations of the buttoned device. For detailed description see reference [45].

**Figure 7 children-07-00111-f007:**
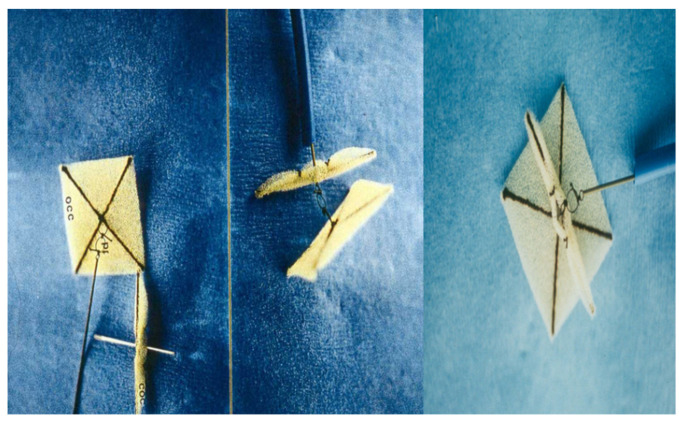
Photograph of a fourth-generation buttoned device. Reproduced from reference [24].

**Figure 8 children-07-00111-f008:**
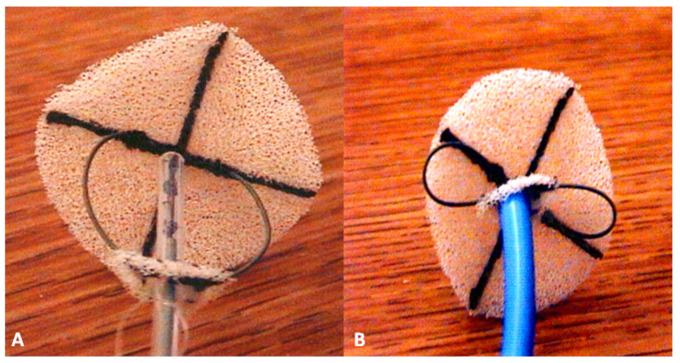
Photograph of a centering-on-demand modification of the buttoned device demonstrating open centering mechanism (**A**) and closed centering mechanism (**B**). Reproduced from Reference [14].

**Figure 9 children-07-00111-f009:**
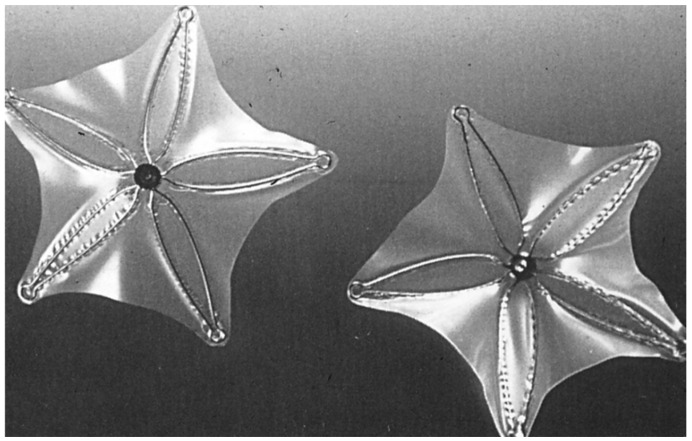
Photograph of the ASDOS (atrial septal defect occluding system). Reproduced from reference [24].

**Figure 10 children-07-00111-f010:**
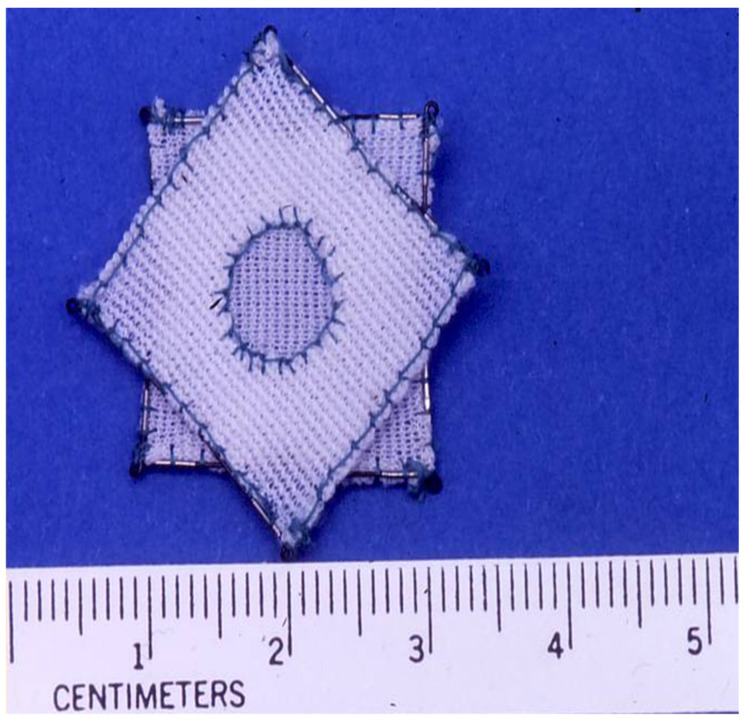
Photograph of a Das Angel Wing device. Reproduced from reference [24].

**Figure 11 children-07-00111-f011:**
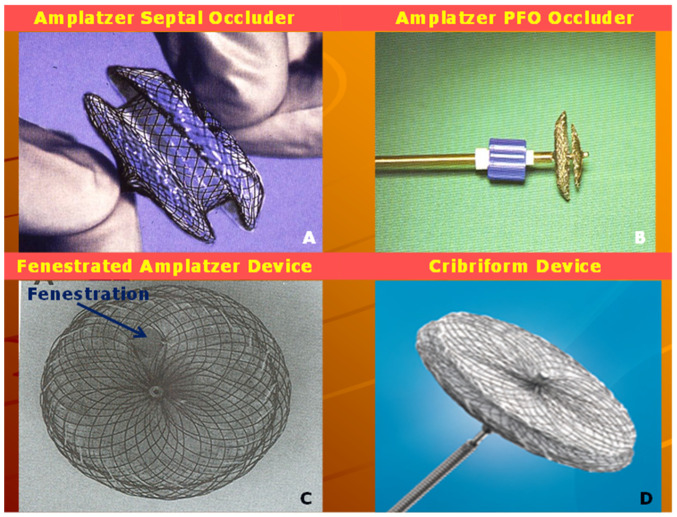
Photographs of Amplatzer Septal Occluder (ASO) (**A**), Amplatzer patent foramen ovale (PFO) occluder (**B**), fenestrated Amplatzer device (**C**) and Cribriform Device (**D**) are shown. Reproduced from Reference [14].

**Figure 12 children-07-00111-f012:**
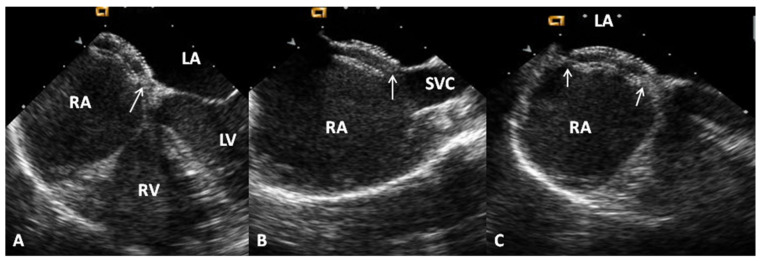
Selected video frames from a trans-esophageal echocardiogram following the implantation of an Amplatzer Septal Occluder to occlude an atrial septal defect, demonstrating the position of both discs in four chamber (**A**), bi-caval (**B**) and long axis (**C**) views. Note that the rims of the defect (thin arrows) are sandwiched between the left atrial (LA) and right atrial (RA) discs. LV, left ventricle; RV, right ventricle; SVC, superior vena cava. Reproduced from Reference [20].

**Figure 13 children-07-00111-f013:**
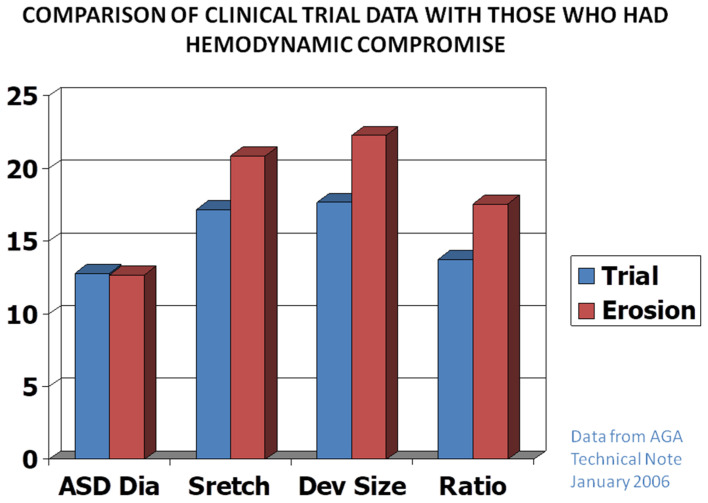
Bar diagram demonstrating the relationships of the data on sizes of atrial septal defects (ASDs) and device sizes between subjects who had no perforation (Trial) and those who had perforation (Erosion). The diameter (Dia) of the atrial defect was same in both groups (left panel); however, the stretched diameter (Stretch), size of the device (Dev size) and the ratio of device to ASD (Ratio) were bigger (middle and right panels) in the subjects who experienced perforation than those who did not. On the basis of these data, it was recommended that devices no larger than 1.5 X the ASD size should be used. Reproduced from reference [88].

**Figure 14 children-07-00111-f014:**
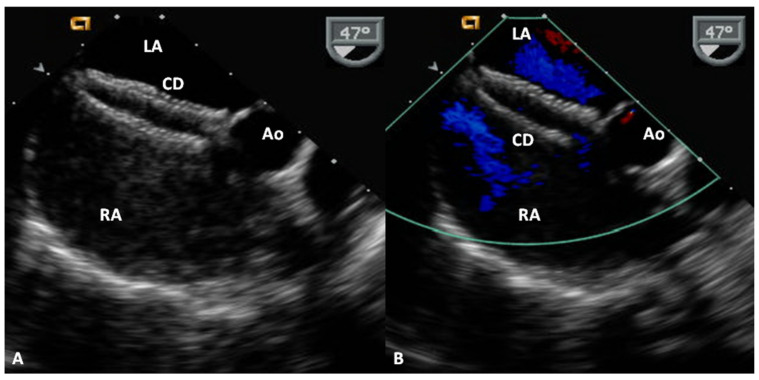
Trans-esophageal echocardiographic study showing Amplatzer Cribriform Device (CD) implanted across a fenestrated atrial septal defect. Note good position of the device (**A**) with no residual shunt (**B**). Ao, aorta; LA, left atrium; RA, right atrium. Reproduced from Reference [20].

**Figure 15 children-07-00111-f015:**
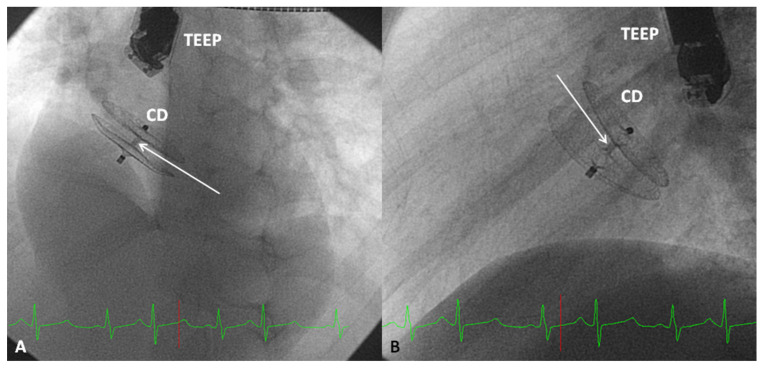
Selected frames from cineflurograms in two different views (**A**,**B**) showing the position of the Amplatzer Cribriform Device (CD) implanted to close a fenestrated atrial defect in the patient shown in Figure 13. Note the thin waist (arrows in A and B) connecting the discs on either side of the atrial septum. While the mechanism for occlusion of atrial septal defect by the Amplatzer Septal Occluder is by the stenting of the atrial septal defect with the waist of the device, the mechanism for the occlusion of a fenestrated atrial septum by the CD is by covering of the defect with the discs of the CD. TEEP, trans-esophageal echocardiography probe. Reproduced from Reference [20].

**Figure 16 children-07-00111-f016:**
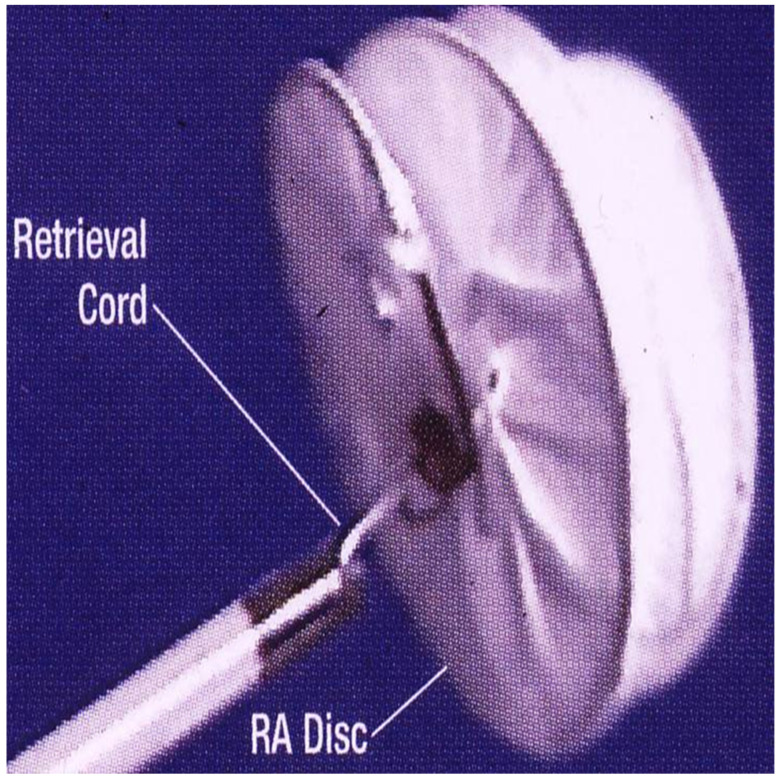
Photograph of a HELEX septal occluder. Reproduced from reference [24]

**Figure 17 children-07-00111-f017:**
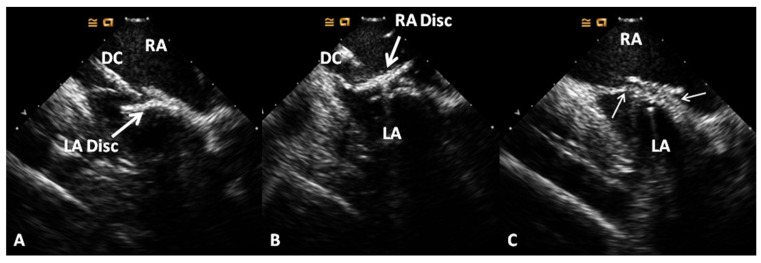
Selected two-dimensional video frames from an intra-cardiac echocardiographic study during HELEX device implantation showing delivery of the left atrial disc (LA Disc) into the left atrium (**A**) and right atrial disc (RA Disc) into the right atrium (**B**). Following disconnection of delivery catheter (DC) from the device (**C**) the rims of the defect (thin arrows) are sandwiched between LA and RA discs. Reproduced from Reference [20].

**Figure 18 children-07-00111-f018:**
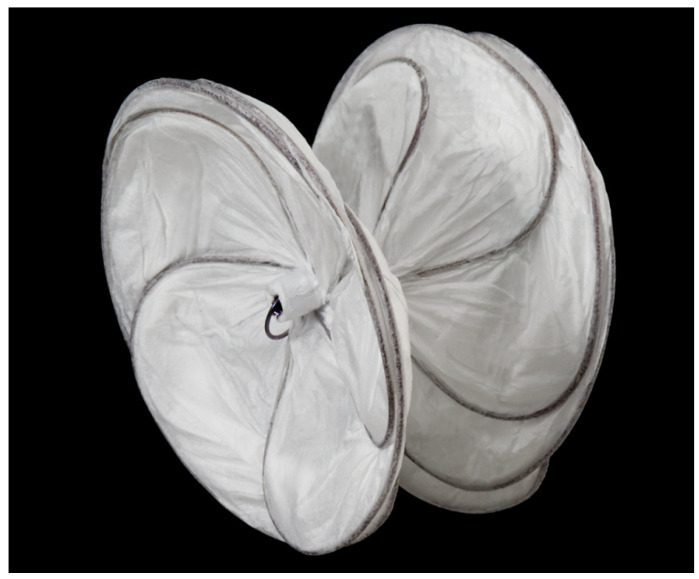
Photograph of a GORE^®^ CARDIOFORM ASD Occluder.

**Figure 19 children-07-00111-f019:**
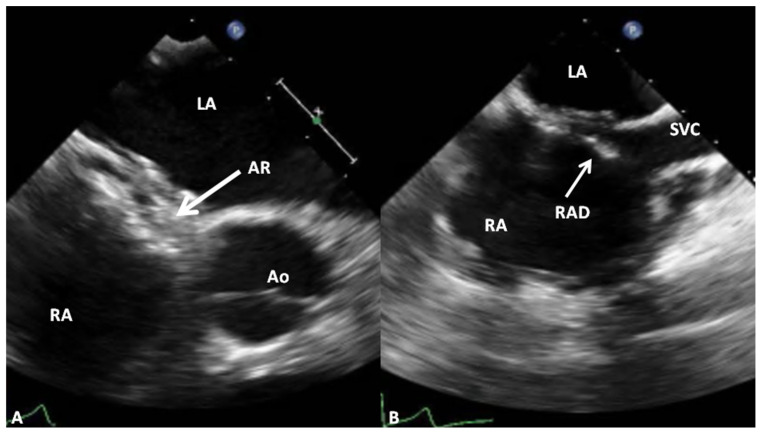
Selected two-dimensional video frames from trans-esophageal echocardiographic study in short axis (**A**) and bicaval (**B**) views demonstrating the position of the GORE^®^ CARDIOFORM ASD Occluder device following its implantation. The aortic rim (AR) (arrow in (**A**)) is seen sandwiched between the right and left atrial discs (**A**). In the bicaval view (**B**) there was some splaying of the superior portion of the right atrial disc (RAD). Ao, aorta; LA, left atrium; RA, right atrium; SVC, superior vena cava.

**Table 1 children-07-00111-t001:** Summary of FDA-approved Devices for Occlusion of secundum ASDs—Part I.

Devices/Studies	Sample Selection	Type of Study	Number of Patients	Age (Years)	Size of Delivery Sheath
**Amplatzer Septal Occluder ***					
Masura et al. [61]	Random sample	Cohort	30	Median-6.1	7F
Hartas et al. [84]	Random sample	Cohort	65	Mean-7.9	7F
**Amplatzer Cribriform Device #**					
Zanchetta et al. [90]	Random sample	Cohort	13	All adults	6 to 7 F
Numan et al. [91]	Random sample	Cohort	16	Median-12.5	6 to 7 F
**Gore HELEX^®^ Device !**					
Jones et al. [104]	Random sample	Cohort	143	Mean-14	8F
Rhodes et al. [105]	Pivotal, continued access and post-approval cases	Cohort	435	Median-6.5	8F
**GORE^®^ CARDIOFORM ASD Occluder @**					
Nyboe et al. [106]	Random sample	Cohort	22	Mean-25.8	10F
Gillespie et al. [112]	Pivotal and continued access	Cohort	400	Median-6.9	10F

* Results of studies in children [62,63,64,65,66,67,68,69,70] and adults [61,64,71,72,73,74,75,76,77,78,79] including FDA-approved US trial [80] and multi-institutional European studies [70,81] were reviewed and tabulated elsewhere [20,21,47,82]; # Similar results [92,93] were seen in other studies; ! Studies in children and adults [102,103,106] were similar; @ Similar results were reported in single institutional [107,108] and Canadian [109], European [110,111] and US [112] multi-institutional studies.

**Table 2 children-07-00111-t002:** Summary of FDA-approved Devices for Occlusion of secundum ASDs—Part II.

Devices/Studies	Size of the ASD (mm)	Implantation Feasibility	Complete Closure at Implantation	Complications/Adverse Events	Complete Closure at 6 to 12 mo after Implantation
**Amplatzer Septal Occluder ***					
Masura et al. [61]	TEE −5 to 21 BS −7 to 19	100%	57%	None	80%
Hartas et al. [84]	TEE −6 to 30 BS −10 to 26	94%	90%	1.6%	98%
**Amplatzer Cribriform Device #**					
Zanchetta et al. [90]	NA	100%	67%	None	96%
Numan et al. [91]	NA	81%	77%	None	92%
**Gore HELEX ^®^ Device !**					
Jones et al. [104]	TEE −13 to 25	88%	73%	5.9%	98%
Rhodes et al. [105]	TEE −1.7 to 25	-	84%	4.8%	98%
**GORE^®^ CARDIOFORM ASD Occluder @**					
Nyboe et al. [106]	TEE −11 ± 0.5	100%	100%	None	100%
Gillespie et al. [112]	TEE −9.7 ± 3 BS −12 ± 3	93.5%	77%	1.7%	86%

BS, Balloon-stretch; NA, Not applicable; TEE, Transesophageal echocardiography; *, #, !, @ Same as Table 1.

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
