# Peer review of "Outcomes of Device Closure of Atrial Septal Defects"

_children, 2020, doi:10.3390/children7090111_

Round 1
Reviewer 1 Report
Manuscript ID:children-886257
Title: Outcomes of Device Closure of Atrial Septal Defects
Minor comments:
- The author has presented a comprehensive and up to date review of the devices available for ASD closure.
- Inclusion of a summary table and discussion of the use cases for each device would have been helpful.
- The manuscript only suffers from minor spelling and grammatical errors.
Author Response
- The reviewer states that the author has presented a comprehensive and up to date review of the devices available for ASD closure and the author thanks the reviewer for this assessment.
- The reviewer suggests Inclusion of a summary table and discussion of the use cases for each device would have been helpful. A Summary Table is included in the revised manuscript as recommended by the reviewer. Presentation of cases with each device is impractical, but for each device, examples of device closure were included in the revised manuscript
- The manuscript only suffers from minor spelling and grammatical errors. I checked the spelling and grammar and made corrections where deemed appropriate.
Reviewer 2 Report
The paper proposes a review of different methodologies to treat atrial septal defect, either or not approved by FDA. I have three main reservations over its present state that need to be addressed before I can advise publication.
- In particular for devices approved and used in human care, the difference between studies are not considered. The sample selection (random sample, multistage sampling, quote sampling…) and type of study (follow-up, cohort, cross-sectional) may influence the analysis that can be performed on the outcomes.
- The advantages and, in particular, limitations of the studies are not mentioned. The latter are very important to understand the results and appreciate the reliability of the study.
- The author listed several approaches but nothing about comparability: is possible to compare them? For instance considering survival… There are more suitable methodologies based on patient characteristic (age, size of the defects, others diseases…) that can be suggested?
Author Response
- The reviewer suggests use of the sample selection (random sample, multistage sampling, quote sampling…) and type of study (follow-up, cohort, cross-sectional) for devices approved and used in humans. This information is now included in a new table created as suggested by the first reviewer.
- The reviewer recommends inclusion of advantages and limitations of the studies. These are now included in the revised manuscript.
- The reviewer suggests inclusion of data comparing the devices on the basis of patient characteristic (age, size of the defects, others diseases…). These are now included in the table and the text, as appropriate.